# Assessing Leadership Capacity in the Food System: The Issue Leadership Scale

**DOI:** 10.3390/foods12203746

**Published:** 2023-10-12

**Authors:** Kevan W. Lamm

**Affiliations:** Department of Agricultural Leadership, Education & Communication, University of Georgia, Athens, GA 30606, USA; kl@uga.edu

**Keywords:** leadership capacity, food system professionals, innovative agricultural practices

## Abstract

The food system represents a complex, interrelated, and interdependent network of individuals and organizations. Despite the unique characteristics of the food system domain, there has never been a comprehensive analysis of leadership capacity amongst food system professionals. The present study provides an empirical assessment of food system leadership capacity using many leadership instruments. The study included pilot (*n* = 125) and primary (*n* = 4185) data collection across 27 food system-related leadership development programs. The results validate the Issue Leadership scale, which comprises seven capacity areas: action, change, communication, critical thinking, strategic planning and visioning, interpersonal traits and characteristics, leadership process, and leadership skills. The Issue Leadership scale captures both perception of importance and current skill level. The study results indicate that food system professionals perceived communication and critical thinking, strategic planning and visioning as the most important leadership capacity areas. Furthermore, respondents indicated the highest level of self-perceived skill in critical thinking, strategic planning and visioning, and interpersonal traits and characteristics. Overall, respondents had higher perceived importance levels (*m* = 4.17) than skill (*m* = 3.44), indicating the need for additional leadership capacity development amongst food system professionals. Respondents also indicated similar levels of opinion leadership (*m* = 3.92), servant leadership (*m* = 3.82), transformational leadership (*m* = 3.78), and transactional leadership (*m* = 3.70), providing an additional empirical assessment of food system professional leadership capacity.

## 1. Introduction

The food system is a complex and multifaceted network that encompasses a range of inputs and outputs which are constrained by multiple factors, “Food-supply-chain networks are impacted by diverse factors […] the complexity of these factors poses a significant challenge” [1] (p. 1). For example, Peng et al. [2] identified approximately 13 steps in the rice supply chain from the plant to purchase, storage, processing, transport, storage, and sale. Therefore, the food system’s complexity represents a unique context within which individual actors may include smallholder farmers, commercial scale farmers, input providers such as seed and fertilizer individuals, financial service providers, policymakers, or food system regulators [3].

The study defines the food system as “social-ecological systems, formed of biophysical and social factors linked through feedback mechanisms” [4] (p. 18). Furthermore, the food system is inextricably linked to the “social, economic and political environment that determine how food system activities are performed” (p. 18). The levels, or scale, at which food system interactions and activities occur are novel and core to the system [5]. The fundamental centrality and criticality of the food system to the health [6], resilience [4], and functioning [7] of society is indisputable. However, complex systems operating at multiple scales require unique leadership capacities. For example, Kickbush and Alakija [6], in a commentary regarding the current state of the UN Sustainable Development Goals (SDGs) [8], cite a lack of strong leaders as one of the challenges and reasons for less progress than anticipated when the SDGs were launched initially. However, despite the importance of food systems and the role of leaders within the food system, little empirical research exists examining the nexus of food system leadership.

The difficulty in defining leadership in an all-encompassing manner has been well established (e.g., [9], [10]). However, leaders have several universal attributes: “leaders are agents of change…Leadership occurs when one group member modifies that motivation or competencies of others in the group. Leadership can be conceived as directing the attention of other members to goals and the paths to achieve them” [9] (p. 25). Within this context, several leadership theories exist [9], where the term theory has been used to describe “a set of interrelated propositions or hypotheses that presents an explanation of some phenomenon” [11] (p. 652). However, the distinction between leadership theory and specific leadership capacities, or competencies, is unclear [12]. Theory “serves to summarize existing knowledge, to explain observed events and relationships, and to predict the occurrence of unobserved events and relationships” [11] (p. 16). Competence describes an individual’s ability to complete a task [13]. In this regard, a conceptual schema may require competence for a particular task to have a prerequisite knowledge of theoretical underpinning yet extend beyond knowledge to application [13]. However, there has been a call for leadership research to continue to evolve beyond the dogma of existing theory.

The recommendations in this article go beyond more traditional situational or contingency models to advocate a fuller and more integrative focus that is multilevel, multicomponent, and interdisciplinary and that recognizes that leadership is a function of both the leader and the led and the complexity of the context.[14] (p. 31)

Previous research has sought to establish a taxonomy, theory, or model-based approach to classify leader behaviors, competencies, or outcomes [9]. “Lacking a taxonomy of this sort, it seems unlikely that substantial progress can be made in the construction of leadership development programs and the generation of more effective models for understanding leader performance” [15] (p. 246). Nevertheless, several leadership theories have been examined across a range of contexts. For example, transformational leadership [16] consists of providing a positive role model, articulating a vision, motivating followers to look beyond their self-interests, providing individualized consideration toward followers, intellectually stimulating followers, and setting high-performance expectations for followers [17]. Transactional leadership is another prominent theory focusing on exchanges between leaders and followers, including rewards and punishments [10]. Servant leadership is another central theory focusing on leader behaviors [18]. Servant leadership consists of empowerment, humility, standing back, authenticity, forgiveness, courage, accountability, and stewardship. Additionally, opinion leadership represents an individual’s centrality within social networks of influence [19]. This may include formal or informal leadership positions; however, these individuals are viewed as knowledgeable and trusted sources of information among followers [20]. 

Within the existing literature, there is evidence to indicate the importance and central role of leadership in the future of food systems. For example, Herens et al. [21] identified leadership as an essential characteristic of food system transformation using multi-stakeholder platforms. However, specific leadership capacities were not identified. The Women’s Empowerment in Agriculture Index (WEAI) [22] includes a measure of leadership; however, the specific scale of the leadership is focused on community group involvement and public speaking. Loboguerrero et al. [23] also identified the importance of enabling and empowering local leaders to improve food system functioning. Previous research has also identified the gap between food system leaders and the general public regarding agricultural issues such as water [24]. Nevertheless, despite the unique context of the food system [25] and evidence indicating the need for leadership capacity development among food system leaders [26], there remains a gap in the literature at this intersection.

The purpose of this study was to examine the leadership capacity of professionals engaged in the food system using the Issue Leadership scale. Multiple research objectives drove the study. First, the study sought to establish the validity of the Issue Leadership scale within the food system. Second, the study intended to quantify the importance and needs of Issue Leadership capacity amongst food system professionals. Lastly, the study examined the relationship between Issue Leadership and traditional leadership models amongst food system professionals.

## 2. Materials and Methods

The study objectives and associated research methodology were developed according to recommendations posited by Crocker and Algina [27], Messick [28], and Lamm et al. [29]. The methods included rigorous validation of the Issue Leadership scale from an internal structural [27,28,30] and external structural perspective [31]. A non-experimental online survey research design was used to collect descriptive and correlational research data. 

### 2.1. Population and Sample

The population for the study was individuals who chose to participate in food system leadership development programs patterned on the Kellogg [32] model. A census included any individual who had participated in a leadership development program affiliated with the International Association for Programs of Agricultural Leadership (IAPAL) organization. A non-probability convenience sample was deemed appropriate for the study based on population size [33]. Respondents were limited to those participants whose program directors agreed to participate in the study. Expressly, 27 program directors agreed to participate. Respondents, including for both pilot and primary studies, were predominately male (68.3%), self-identified as white (92.7%), and were between 50 and 59 years old (34.7%). Detailed demographic data of respondents are available in Lamm et al. [34]. 

### 2.2. Instrumentation

A questionnaire was developed to collect data from food system leadership development program participants. Based on recommendations within the literature [27], the Issue Leadership scale was developed based on results from a previous study identifying the primary outcomes associated with food system leadership development programs (see [26]). The scale was designed by the researcher and then reviewed and approved for content and face validity and reliability using a panel of experts representing survey development, evaluation, leadership development programming, and organizational behavior [27,28].

The scale specifically asked respondents to indicate how vital (importance) each competency factor was to success in their current position. General competency factors were established on a five-point Likert-type scale. Responses included: 1—Not at all important, 2–Very Unimportant, 3–Neither Important nor Unimportant, 4—Very Important, and 5—Extremely Important. Next, respondents were asked to indicate how skilled (skill) they believed themselves to be on the provided competency factors. Competency factors were established on a six-point Likert-type scale. Responses included: 0—Not Applicable (no knowledge), 1–Fundamental Awareness (basic knowledge), 2–Novice (limited experience), 3–Intermediate (practical application), 4–Advanced (applied theory), and 5—Expert (recognized authority). Previous research has found that Likert-type scales with differing numbers of items do not tend to vary regarding their internal structure, including means, standard deviations, correlations, Cronbach’s α, or factor loadings [35]. Therefore, an importance scale with five items and a skill scale with six items were deemed acceptable and appropriate given the potential for an individual to have no skill within a factor.

### 2.3. Data Collection

#### 2.3.1. Pilot Study One

Alums from the United States-based Florida Wedgworth Leadership Institute for Agriculture and Natural Resources were surveyed as a pilot of the survey. The questionnaire was administered through the Qualtrics online tool, and all subsequent communications were conducted in adherence to the Tailored Design Method recommendations [36]. First, the program director emailed a pre-notice to all program participants one week before the questionnaire. Second, the researcher followed up with an email invitation to each participant, including a link to the questionnaire and a requested response date three weeks later. Approximately one week after the invitation to the questionnaire was sent, the researcher sent a reminder email to non-responders. Two weeks after the original invitation to the questionnaire, the researcher sent a second reminder email to non-responders. Two days before the questionnaire’s close, the researcher sent a third reminder email to non-responders. The researcher sent a fourth and final email to non-responders on the requested response date. A thank you email was sent to all respondents one week after the closure of the questionnaire.

There were 207 questionnaires sent with 22 invalid email addresses for 185 potential respondents. There were 98 completed questionnaires returned for a 53% response rate. This was considered acceptable according to established social science response rates [37]. Data from pilot one were analyzed to ensure survey validity and reliability. Minor wording modifications to the questionnaire instructions were made based on responses. None of the modifications were germane to the purpose of this research. An expert panel reviewed and approved the revised questionnaire (e.g., [27]).

#### 2.3.2. Pilot Study Two

The revised questionnaire was piloted a second time with the current class from the Florida Wedgworth Leadership Institute for Agriculture and Natural Resources. A second pilot was deemed appropriate due to the modifications associated with the questionnaire, although updates were minor. All subsequent analyses aggregated data related to pilot studies one and two into a single pilot data group referred to as the pilot study.

The questionnaire was again administered through the Qualtrics online tool, and all subsequent communications were conducted in adherence to the Tailored Design Method recommendations [36] described above. Twenty-nine survey invitations were sent out, and 27 surveys were completed for a 93% response rate. Data were analyzed to confirm survey validity and reliability (e.g., [27]).

#### 2.3.3. Primary Study

After completing pilot studies one and two, the researcher contacted the 27 program directors who agreed to participate. The study comprised 25 programs from the United States and 2 from Canada. The questionnaire was administered through the Qualtrics online tool, and all subsequent communications were conducted in adherence to the Tailored Design Method recommendations described above [36].

A total of 8435 invitations were sent, with 943 invalid email addresses for 7152 potential respondents. There were 4185 surveys returned for a 59% response rate. This was considered acceptable according to established social science response rates [37].

### 2.4. Data Analysis

After respondents entered data into the online tool, results were downloaded and analyzed using the Statistical Package for the Social Sciences (SPSS) version 21 and Mplus version 7 [38]. Respondent characteristics between pilot studies one and two were analyzed using chi-square statistics. No statistically significant differences were observed between groups related to demographic factors, including gender and age. Additionally, pilot characteristics were analyzed relative to primary study respondents using chi-square statistics. Again, no statistically significant differences between groups were observed based on the demographic characteristics of gender and age. Individual item scores for the Importance and Skill variables were averaged to calculate the Issue Leadership scale score. Descriptive statistics were calculated to determine the level of importance and self-perceived skill level that food system leadership development program participants ascribed to the general competency factors, and respondents’ overall leadership competence scores [11]. Based on recommendations from Ferguson and Cox [39], result distributions were further analyzed for skewness and kurtosis. Skewness values less than two and kurtosis values less than seven were considered acceptable, given established thresholds for factor analysis within psychological research [40,41].

Internal structure validity of the Importance and Skill latent variables was analyzed using descriptive statistics, Cronbach’s α, exploratory factor analysis (EFA), and confirmatory factor analysis (CFA). Cronbach’s α was calculated across the resulting latent variables to determine item internal consistency. Based on established social science standards, an observed value of 0.70 or above was considered acceptable [42,43,44]. Next, an EFA was completed using a Promax rotation in SPSS version 21 to examine the hypothesized single latent variable within the pilot study respondent groups, including pilot studies one and two. Eigenvalues were used to determine the factor structure of each latent variable. Lastly, a CFA was calculated using Mplus version 7 [38] to investigate the factor structure of the latent variables. 

External structure validity was analyzed using correlations. Hypothesized relationships between the Issue Leadership latent variables (Importance and Skill) were examined relative to previously established constructs within the nomological network of similar concepts [28,31], specifically, transformational leadership measured using the Transformational Leadership Inventory [17], transactional leadership [45], servant leadership using the Van Dierendonck and Nuijten [18] Servant Leadership Survey, and opinion leadership using a scale designed by Childers [46]. All scales were five-point Likert-type and included the following options: 1—Strongly Disagree, 2—Disagree, 3—Neither Agree nor Disagree, 4—Agree, 5—Strongly Agree. Correlation coefficients were calculated to determine the magnitude of the relationship between latent variables and external scale factors. Results were evaluated based on Davis’ [47] convention.

## 3. Results

### 3.1. Descriptive Statistics

Descriptive statistics were calculated to examine the response distributions across the two latent variables in both the pilot study and the primary study and ensure all items fell within an acceptable range of responses before conducting subsequent analyses [30].

#### 3.1.1. Importance Variable Descriptive Results

The response distribution for items in the Importance latent variable for the pilot study respondents are presented in Table 1, and the primary study responses are shown in Table 2. Most respondents indicated that items were very important or extremely important within the pilot and primary study conditions. Less than 2% of respondents indicated that any item was not at all important. Within both the pilot and primary studies, the data were skewed, indicating respondents generally placed a higher value on the importance of the items than a normal distribution of responses would predict. However, all observed values fell within an acceptable range of normality [40]. Cronbach’s α was calculated for the Importance variable within the pilot and primary studies (Table 3). Within both the pilot and primary studies, Cronbach’s α values over 0.80 were observed, indicating acceptable internal consistency.

#### 3.1.2. Skill Variable Descriptive Results

The response distribution for items in the Skill latent variable for the pilot study respondents and primary study respondents are presented in Table 4 and Table 5. Within both the pilot and primary study conditions, most respondents indicated their skill level was either intermediate, meaning a practical application of the item, or advanced, meaning they could apply the theory as appropriate. Result distributions were further analyzed for skewness and kurtosis; within the pilot and primary studies, the data were within acceptable ranges. Cronbach’s α was calculated for the Skill variable within the pilot and primary studies (Table 6). Within both the pilot and primary studies, Cronbach’s α values were over 0.80, indicating acceptable internal consistency.

### 3.2. Exploratory Factor Analysis

Exploratory factor analysis (EFA) was conducted using the pilot study responses. After establishing the sufficiency of the respondent-to-item ratio [39], the Kaiser–Meyer–Olkin test of sampling adequacy and Bartlett’s test of sphericity were employed to ensure that the observed factor structures were not found by chance. The Kaiser–Meyer–Olkin test yielded a value of 0.84 for the Importance latent variable and 0.82 for the Skill latent variable. Both values are acceptable given a minimum threshold of 0.5, with scores above 0.8 considered very good [48]. Additionally, the results of Bartlett’s test of sphericity were all significant, indicating the psychometric adequacy of the samples [48]. Results of the Kaiser–Meyer–Olkin test and Bartlett’s test of sphericity indicated that the data were sufficient to determine whether a stable factor structure existed within the data. 

To investigate factor structure within the data, Kaiser’s [49] eigenvalue greater than 1 (K1) was employed as recommended by the literature [50]. Based on the K1 analysis, a single factor with an eigenvalue more significant than one was identified within the Importance and Skill latent variables, indicating the unidimensional nature of the data (Table 7). Overall, the results of the EFA analysis indicated that a single, stable factor existed within the Importance latent variable and Skill latent variable.

### 3.3. Confirmatory Factor Analysis

Based on the results of the EFA, specifically, that a single stable factor was observed within the Importance latent variable and Skill latent variable, the hypothesized factor structure was further analyzed through CFA. The responses collected from the primary study were included in the CFA.

The first model examined included all seven items loading onto a single latent variable. Error terms amongst items were treated as uncorrelated. A representation of the hypothesized model is presented in Figure 1. Model fit statistics were calculated and included the Chi-square test of model fit, root mean square error of approximation (RMSEA), comparative fit index (CFI), Tucker Lewis Index (TLI), and standardized root mean square residual (SRMR). Model fit results are presented in Table 8.

According to Hu and Bentler [51], several benchmarks have been established to analyze model fit statistics and thus identify model misspecification. Within Model 1, the observed SRMR values for all latent variables indicated good model fit; however, RMSEA and TLI values were not within acceptable ranges. Consequently, results from Model 1 indicated there may be a model misfit. Based on recommendations within the literature [52], CFA Model 2 was modified based on the potential for response confounding due to item construction. Specifically, items may correlate due to respondent relative answering based on item or stem preconditioning [53]. In Model 2, based on structural similarities, the disturbance terms for the leadership process and leadership skill items were allowed to correlate (Figure 2). Model 2 fit statistics are presented in Table 9.

The fit statistics associated with Model 2 indicate that the model fits the data. All fit statistics ranged from acceptable to very good [51]. The standardized solution of the single-factor structures for each latent variable is reported in Table 10. Results indicated that all items load on a single factor.

### 3.4. External Structure Validity and Standard Leadership Measures

External data validity was collected from volunteer respondents from the primary study. Specifically, at the conclusion of the evaluation survey in which the study was embedded, respondents were allowed to continue onto a secondary set of research and scale-based questions. A total of 992 individuals completed all additional measures, including transformational leadership, transactional leadership, servant leadership, and opinion leadership. Correlations were calculated to determine the directionality and magnitude of the relationship between factors (Table 11). 

The Skill latent variable had a positive, statistically significant correlation with all other measures and factors. A moderate relationship was observed between Skill and transformational leadership and servant leadership. Correlations ranged from low to moderate across all measures. The Importance latent variable had a positive and statistically significant relationship with all measures and factors. Overall, correlations ranged from negligible to low in magnitude. Between Importance and Skill (0.36) a moderate, positive relationship was observed. This observation would indicate no multicollinearity between Issue Leadership Skill and Importance latent variables.

## 4. Discussion

Although numerous theories of leadership are available within the literature, there has been little consensus as to an all-inclusive model [9]. Furthermore, one of the challenges associated with leadership theories has been the limited number of empirically developed measures which provide valid and reliable data; frequently, models are proposed based on anecdotal experience and lack rigorous design and validation [54]. Additionally, there are no known leadership models which are specifically focused on the food system and the unique leadership characteristics associated with the domain [26]. The Issue Leadership framework, composed of both the Importance and Skill latent variables, provides a praxis forward model to assess leadership capacity within the food system. The absence of a robust and theoretically grounded food systems leadership framework has limited the development of standard leadership capacity criteria within the food system.

The present study validates the Issue Leadership scale with a large sample of food system professionals. Additionally, the study results provide novel insights and observations regarding the leadership capacity of the audience. The internal and external structure of the Issue Leadership scale was established using a wide and robust range of validation techniques, as recommended within the literature [27,28]. Therefore, the study’s results provide both a theoretical and practical contribution to the existing food systems literature. 

### 4.1. Theoretical Contributions to Food Systems

The development and validation process associated with the Issue Leadership scale represents a model for future leadership and social science scale development within the food system. The methodical collection and analysis of validity evidence gathered based on Crocker and Algina’s [27] and Messick’s [28] recommendations ensured that the content, response process, internal structure, external structure, and consequential use of the scale were appropriate. For example, the relationship between the Importance and Skill latent variables and transformational leadership [17], transactional leadership [45], servant leadership [18], and opinion leadership [46] was examined.

According to Davis’ [47] convention, the magnitude of the relationship between the Importance latent variable and the external constructs ranged from negligible to low, all in the positive direction. The importance latent variable had the largest observed relationship with the overall transformational (0.28), servant (0.27), and transactional (0.18) index values. The magnitude of the relationship with opinion leadership (0.11) was also low and in the positive direction. 

The observed relationships between the Skill latent variable and previously established constructs ranged from low to moderate in magnitude and were positive. Across all factors and indices, the skill latent variable correlation coefficients were larger than those of the importance latent variable. However, the relationships with the largest magnitude were observed with the overall transformational (0.44), servant (0.40), and transactional (0.27) index values. The magnitude of the relationship with opinion leadership (0.17) was also low and in the positive direction.

Overall, results were consistent with the magnitudes and directionality hypothesized prior to analysis. Specifically, both Importance and Skill were expected to have positive relationships with existing constructs, which is consistent with what was observed. Consequently, it was determined that the proposed latent variables were sufficiently related to, yet non-redundant with, previously established leadership constructs; thus, providing evidence of external structure validity, and importantly, a novel measure of leadership capacity specifically developed for use within the food system.

There are a number of leadership measures available such as the Multi-Factor Leadership Questionnaire [16], which provides a measure of transformational leadership, or the Servant Leadership Survey [18]; however, there are few, if any, which provide a food system-specific assessment of leadership capacity. Identifying the behaviors of food system leaders first (see [26]) allowed the Issue Leadership scale to be developed free from the constraints of existing models. Therefore, the resulting scale has much more flexibility to conform to the needs of food system leaders, rather than leaders needing to conform to theory. 

### 4.2. Recommendations and Implications for Future Research

A number of future research areas are identified based on study results. One of the primary limitations of the study was that data were collected through self-report. A recommendation for future research would be to have a sample of participants complete the Issue Leadership scales and to have those individuals also rated by knowledgeable external parties. The external raters would complete the Issue Leadership scales relative to their perceptions of the respondent. The intent of the research would be to determine whether there were any statistically significant differences between the self and other ratings. Additionally, similar to previous research with transformational leadership [55], it is recommended that the Issue Leadership scales be analyzed according to various demographic and contextual characteristics. A further refinement of Issue Leadership capacity trends among different audiences within the food system would provide greater individualization of capacity-building efforts. 

Although the Issue Leadership scales were analyzed against the Transformational Leadership Inventory [17], the Servant Leadership Survey [18], and opinion leadership [46] it is recommended that additional constructs be analyzed relative to Issue Leadership scales. For example, examining the Issue Leadership scales against personality [56] may illuminate and clarify relationships. Furthermore, an analysis of the Issue Leadership scales and actual leadership role activity warrants further investigation. Specifically, whether levels of leadership capacity are predictive of leadership roles, or the types of roles assumed within the food system.

Lastly, although the Issue Leadership scales have demonstrated great promise within the food system domain, the primary population for the study was existing food system professionals. An investigation of the transferability of the scale beyond the current population is suggested. For example, the validity of the scale with undergraduate, or pre-professional, populations is also suggested. Establishing the validity of the Issue Leadership scales with a more diverse audience may have implications for leadership capacity development in the food system more broadly.

### 4.3. Practical Contributions to Food Systems

From a practical perspective, this study provides the first, and largest, evaluation of leadership capacity within the food system within the academic literature. For example, within the primary study, respondents identified Communication as the most important area of leadership within the food system domain, closely followed by Critical Thinking, Strategic Planning and Visioning. These descriptive findings are noteworthy as they provide a quantitative comparison between core areas of leadership. It is also interesting to note that change was identified as the least important area of leadership. The overall Issue Leadership Importance index score was relatively high (*M* = 4.17) indicating the perceived importance of Issue Leadership capacity within the food system. Additionally, this value represents a benchmark to examine and compare future Issue Leadership Importance scale scores, and to examine potential differences within and between groups. 

From an Issue Leadership Skill perspective, there are also numerous observations that provide practical insights regarding food system leadership capacity. For example, in the primary study Critical Thinking, Strategic Planning and Visioning was identified as the area respondents perceived themselves to have the highest level of skill closely followed by Interpersonal Traits and Characteristics. Overall, respondents had a slightly lower perception of their skill related to Issue Leadership (*M* = 3.44) relative to importance.

Based on a review of the literature, this is the first empirically developed leadership scale that not only provides a measure of skill, but also importance. The study results address one of the main criticisms leveled at leadership development programs: that leadership is presumed to be universal and appropriately agnostic of situation [54]. Without some measure of perceived importance, the skill value associated with the proposed scale would provide similar information to existing scales within the literature. A practical contribution of the Issue Leadership scale over existing measures of leadership is the acknowledgement of context in the form of importance. Although there has been post hoc analysis of leadership effectiveness given certain antecedent conditions, such as gender or organizational context and level of transformational leadership [55], there is a lack of individually identified attribution between the leadership measure and the outcome. Providing food system actors and leaders with direct and quantifiable insights as to the importance of leadership capacities should help to establish a common lexicon of food system leadership, leadership strengths, and opportunities for leadership development. The importance aspect also represents an objective benchmark to evaluate results across the food system.

The Importance and Skill results together represent a novel, practical contribution to the food system. For example, the Communication area had the third highest level of perceived skill; meanwhile, Communication was the top area as it relates to Importance. The synthesis of these results, across all areas, has the potential to impact future food system leadership capacity priority foci. The Critical Thinking, Strategic Planning and Visioning area was scored high in both the Importance and Skill indices, indicating there is likely a balance between perceived need and capacity. To the contrary, the difference in absolute scores related to Communication indicate a potential gap, and a likely area for capacity building need. Lastly, Change had the lowest level of perceived importance and skill relative to the other areas. Again, this finding provides a practical perspective on what food system leadership capacity areas are not only strong candidates for development (perceptions of skill) but also prioritization (perceptions of importance).

This study also provides empirical benchmarks for food system leadership capacity across multiple established measures of leadership. Specifically, respondents had the highest observed mean score in opinion leadership [46], followed by servant leadership [18], transformational leadership [17], and transactional leadership [45]. The observed scores across all scales indicate food system leaders have self-reported leadership capacity across a range of leadership models, with little practical difference observed between scales.

### 4.4. Recommendations and Implications for Practice

The group most proximal to the study, and consequently best positioned to benefit immediately, is educators and professionals focused on food system leadership capacity development. One of the primary areas where educators will benefit from using the Issue Leadership scales is to gain insights into food system professionals, both actual and potential. Targeting specific capacity needs will help to improve the efficiency and effectiveness of capacity-building efforts. Additionally, educators can use the Issue Leadership scales to measure leadership capacity development impact. Unlike other leadership measures, Issue Leadership provides a control mechanism for respondent context and the perceived importance of capacity areas. A recommendation would be to use the Issue Leadership scales to conduct true pre-test and post-test measures to objectively evaluate whether intervention (capacity building) efforts were impactful. An additional recommendation is to use Importance as a control variable to predict how Skill may impact dependent variables of interest, such as work performance. Another recommendation is for food system leadership capacity development programs to modify and adapt efforts based on observed trends. For example, reallocating efforts and resources from Critical Thinking, Strategic Planning and Visioning to Communication capacity building may result in more consequential impacts amongst food system professionals. This recommendation enables educators to provide the most appropriate learning experiences possible where capacity-building efforts are tailored to the learner’s needs.

In addition to professionals focused on food system leadership capacity development, food system professionals themselves are also recommended to use Issue Leadership scale results to gain insights into their current leadership capacity and to analyze their leadership development goals critically. Self-reflection and personal insights are essential leadership capacity building techniques [9]. A recommendation is for food system professionals to use results to identify where observations are consistent with expectations and where observations may indicate an opportunity for further development. As food systems continue to evolve, associated professionals will need to address multiple mandates. For example, precision farming practices [57] are innovative agricultural practices that are intended to improve sustainability and efficiency. However, the innovations also may require new skills and capacities, “We further suggest training programs addressing the development of advanced competencies needed to promote agricultural innovation” (p. 11). Furthermore, efforts to produce more nutritious foods in a more inclusive environment also represent food system trends requiring leadership capacity development. Within an edible mushroom context, recent research has proposed, “These implications highlight the importance of funding and training a new generation of young mushroom growers who can successfully expand fungiculture” [58] (p. 15).

### 4.5. Limitations

Despite the theoretical and practical contributions of the present study, there are several limitations which must be addressed. First, respondents were limited to individuals associated with the International Association for Programs of Agricultural Leadership (IAPAL) organization, “a consortium of leadership programs in the USA and several other countries” ([59], para. 1). As leadership capacity building programs, the participants involved in the programs may not be representative of all food system professionals. Although the current study is one of the most comprehensive in the existing literature, interpretation of results should be limited to study respondents and not generalized to the broader population. 

A second main limitation is interpreting Skill and Importance scale scores based on different underlying scale options. Specifically, Importance ranged from 1 to 5 whereas Skill ranged from 0 to 5. From a practical perspective, only 9 out of 3489 responded with a 0 in the Skill area, indicating a statistically insignificant contribution to the findings (0.26%); however, caution is still recommended in making lateral comparisons between overall Importance and Skill index scores. A recommendation for future studies would be to harmonize scores on a standard five-point Likert-type scale and thus improve interpretability.

Lastly, results from the validation process show that the similarity between Leadership Skill and Leadership Process areas may indicate respondent relative answering based on item or stem preconditioning [53]. An associated recommendation would be to rename the areas with cognitively distinct labels to represent the underlying leadership area in future research.

## 5. Conclusions

The food system is a complex, interrelated network [4] which operates at multiple scales [5]. Previous research has proposed a conceptual leadership model within the food system operating at the interpersonal, organizational, community, and policy levels [26]. The proposed Issue Leadership framework (please refer to Appendix A for scale details) provides a novel lens through which to consider leadership within the food system and is directly supportive of identified leadership capacity needs within the food system (e.g., [21,22,23]). However, despite the identified need for leadership capacity, there have been no large-scale, empirical assessments of such capacity amongst food system professionals. The present study provides a robust, praxis and utility forward measure and set of benchmark findings to inform current and future food system leadership capacity development efforts.

## Figures and Tables

**Figure 1 foods-12-03746-f001:**
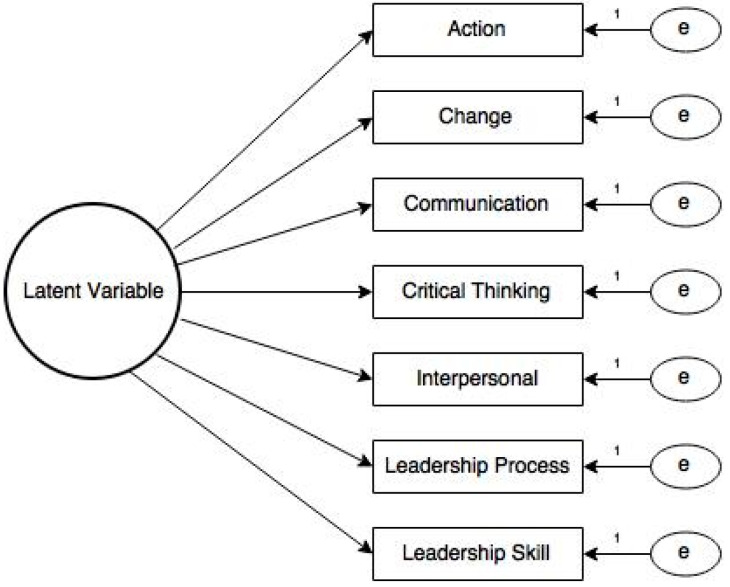
Model 1 hypothesized latent variable model structure. Note: error terms amongst items are treated as uncorrelated.

**Figure 2 foods-12-03746-f002:**
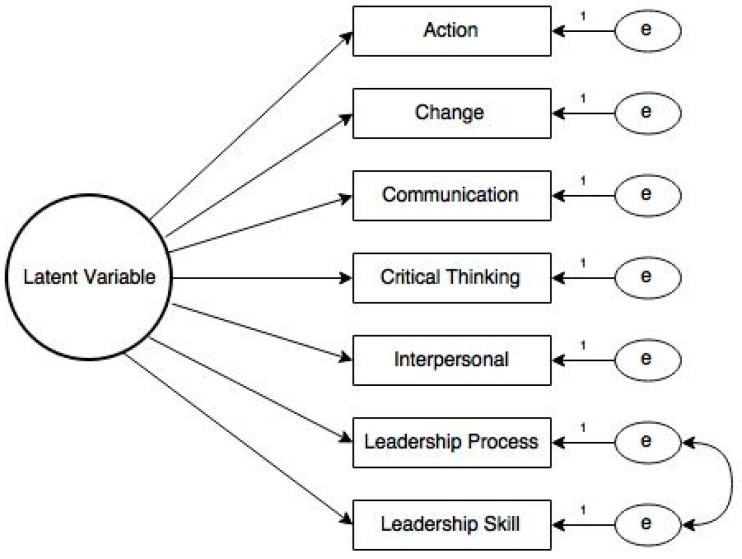
Model 2 hypothesized latent variable model structure. Note: Error terms between Leadership Process and Leadership Skill are allowed to correlate; all other error terms amongst items are treated as uncorrelated.

**Table 1 foods-12-03746-t001:** Importance variable items response distribution pilot study (1 = not at all important; 5 = extremely important).

Item	*n*	1 %	2 %	3 %	4 %	5 %	Skewness	Kurtosis
Critical Thinking, Strategic Planning and Visioning	133	1.5	1.50	3.76	33.08	60.15	−2.10	5.85
Communication	133	0.00	4.51	4.51	44.36	46.62	−1.26	1.75
Interpersonal Traits and Characteristics	133	0.75	3.76	5.26	48.12	42.11	−1.37	2.68
Leadership Process	131	0.76	1.53	7.63	50.38	39.69	−1.18	2.76
Action	133	0.75	3.01	7.52	57.89	30.83	−1.15	2.71
Leadership Skill	132	1.52	1.52	14.39	54.55	28.03	−1.06	2.39
Change	131	0.76	2.29	25.19	58.02	13.74	−0.58	1.27

Note: 0—Not Applicable (no knowledge); 1—Fundamental Awareness (basic knowledge); 2—Novice (limited experience); 3—Intermediate (practical application); 4—Advanced (applied theory); 5—Expert (recognized authority).

**Table 2 foods-12-03746-t002:** Importance variable items response distribution primary study (1 = not at all important; 5 = extremely important).

Item	*n*	1 %	2 %	3 %	4 %	5 %	Skewness	Kurtosis
Communication	3517	0.74	1.79	6.54	39.32	51.61	−1.46	2.92
Critical Thinking, Strategic Planning and Visioning	3539	0.96	1.70	4.80	42.19	50.35	−1.59	3.87
Interpersonal Traits and Characteristics	3539	0.73	2.18	10.51	48.01	38.57	−1.06	1.72
Leadership Process	3530	0.62	2.35	9.29	51.64	36.09	−1.04	1.89
Leadership Skill	3546	0.54	3.02	14.3	53.02	29.13	−0.79	1.02
Action	3552	0.28	2.65	11.15	59.23	26.69	−0.77	1.42
Change	3517	0.65	3.38	25.82	53.45	16.69	−0.48	0.56

**Table 3 foods-12-03746-t003:** Importance latent variable pilot and primary study descriptive statistics.

Study	*n*	Min	Max	Mean	S.D.	α
Pilot	128	1.71	5.00	4.21	0.54	0.84
Primary	3401	1.00	5.00	4.17	0.51	0.81

**Table 4 foods-12-03746-t004:** Skill variable items response distribution pilot study (1 = not at all important; 5 = extremely important).

Item	*n*	0 %	1 %	2 %	3 %	4 %	5 %	Skewness	Kurtosis
Critical Thinking, Strategic Planning and Visioning	130	0.00	0.00	3.85	30.77	50.00	15.38	−0.14	−0.31
Leadership Process	129	0.00	0.00	6.20	30.23	49.61	13.95	−0.24	−0.26
Action	131	0.00	0.00	5.34	44.27	37.4	12.98	0.22	−0.49
Interpersonal Traits and Characteristics	131	0.00	1.53	1.53	32.06	51.91	12.98	−0.56	1.35
Leadership Skill	130	0.00	2.31	7.69	36.92	44.62	8.46	−0.52	0.54
Communication	131	0.00	1.53	10.69	36.64	43.51	7.63	−0.39	0.10
Change	130	0.00	3.08	13.08	41.54	36.92	5.38	−0.38	0.12

**Table 5 foods-12-03746-t005:** Skill variable items response distribution primary study (1 = not at all important; 5 = extremely important).

Item	*n*	0 %	1 %	2 %	3 %	4 %	5 %	Skewness	Kurtosis
Critical Thinking, Strategic Planning and Visioning	3484	0.20	1.46	7.55	35.45	42.62	12.72	−0.44	0.46
Interpersonal Traits and Characteristics	3489	0.26	1.75	5.50	34.82	45.17	12.50	−0.58	0.96
Communication	3472	0.12	1.90	7.69	40.50	40.47	9.33	−0.38	0.50
Action	3498	0.14	1.46	5.23	45.57	38.28	9.32	−0.25	0.76
Leadership Process	3485	0.23	1.61	9.47	40.26	39.91	8.52	−0.39	0.48
Leadership Skill	3487	0.23	2.47	10.73	42.36	36.71	7.51	−0.38	0.44
Change	3466	0.23	3.46	14.74	46.91	29.14	5.51	−0.26	0.30

**Table 6 foods-12-03746-t006:** Skill latent variable pilot and primary study descriptive statistics.

Study	*n*	Min	Max	Mean	S.D.	α
Pilot	127	2.00	5.00	3.56	0.57	0.83
Primary	3388	0.00	5.00	3.44	0.61	0.84

**Table 7 foods-12-03746-t007:** EFA K1 analysis.

Latent Variable	Eigenvalue	% of Variance Explained
Importance	3.56	42.96
Skill	3.48	41.50

**Table 8 foods-12-03746-t008:** Model 1 CFA fit statistics.

Latent Variable	Χ^2^	df	*p*	RMSEA	CFI	TLI	SRMR
Importance	513.10	14	0.00	0.10	0.92	0.88	0.04
Skill	619.34	14	0.00	0.11	0.93	0.89	0.04

**Table 9 foods-12-03746-t009:** Model 2 CFA fit statistics.

Latent Variable	Χ^2^	df	*p*	RMSEA	CFI	TLI	SRMR
Importance	317.95	13	0.00	0.08	0.95	0.92	0.04
Skill	382.76	13	0.00	0.09	0.96	0.93	0.04

**Table 10 foods-12-03746-t010:** Standardized factor loadings of CFA.

Item	Importance	Skill
Action	0.58	0.69
Change	0.52	0.62
Communication	0.61	0.59
Critical Thinking, Strategic Planning and Visioning	0.64	0.65
Interpersonal Traits and Characteristics	0.61	0.63
Leadership Process	0.65	0.70
Leadership Skill	0.62	0.67

**Table 11 foods-12-03746-t011:** Intercorrelations among latent variables, opinion, transformational, transactional, and servant leadership.

Item	*M*	S.D.	1	2	3	4	5	6
1. Issue Leadership Importance Variable	4.17	0.51	(0.81)					
2. Issue Leadership Skill Variable	3.44	0.61	0.36	(0.84)				
3. Opinion Leadership	3.92	0.77	0.11	0.17	(0.91)			
4. Transformational Leadership	3.78	0.33	0.28	0.44	0.22	(0.75)		
5. Transactional Leadership	3.70	0.46	0.18	0.27	0.10	0.42	(0.75)	
6. Servant Leadership	3.82	0.30	0.27	0.40	0.20	0.65	0.47	(0.86)

Note: Observed Cronbach’s α values are presented on the diagonal. All values are significant at *p* < 0.001.

## Data Availability

The data are not publicly available due to confidentiality restrictions. Please reach out to the corresponding author for questions related to data availability.

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
