# Peer review of "Assessing Leadership Capacity in the Food System: The Issue Leadership Scale"

_foods, 2023, doi:10.3390/foods12203746_

Round 1

Reviewer 1 Report

Comments and Suggestions for Authors

This is an interesting paper focusing on the assessment of leadership capacity in the food system.

From the abstract, some salient findings of this study were conspicuously missing. Provide some results data from your study in this section.

From the introduction, there is a need for authors to present a more robust background information on this study. The concept of food system was not  presented as expected. If there is no concept of food system, there will not be leadership capacity in the food system. Present a robust background information on the food system in recent time. Are there previous works on leadership capacity in the food system in literature? Kindly, present a brief background information on this. 

Incorporate 1.1 section into the introduction properly without making it a subheading. Do not itemise your objectives. Kindly, present them in a flowing sentences. 

It is important for the author to clearly identify the gap(s) in knowledge this study sets to fill. This can be done by identifying other studies (if any) and revealing the gap(s) in knowledge concerning this study.

Which type of sampling method did you use for this study?

Where is the location of this study, study area - country? 

Add country to "Florida Wedgworth leadership institute for Agriculture and natural Resources.

Comments on the Quality of English Language

Fine

Reviewer 2 Report

Comments and Suggestions for Authors

The paper was revised according to the journal rules.

Few revisions are required and they are reported below:

- I suggest to revise the abstract adding the crucial results achieved

- please add the nomenclature list for all acronyms and parameters

- the introduction section can be improved, clarifying the state-of-the-art gap, and adding more references to follow

- figure 1 and 2 should be revised

- please use the same style for the references

- please revise section 2, and check that all features and details are properly reported

- please check and revise "...less than .26% of ..."

- I suggest to revise the conclusion section

Comments on the Quality of English Language

minor comments

Reviewer 3 Report

Comments and Suggestions for Authors

The author describes findings from a well designed study to advance the understanding of food system professional leadership capacity. The Issue Leadership scale was validated in two pilot and one primary study of members of the International Association for Programs of Agricultural Leadership (IAPAL) organization.

The background, methods, results and discussion are thorough.

Suggestions:

Include in the abstract information about the survey and survey respondents.

Include in Supplemental Materials the questionnaire.

Restructure the sentence (perhaps separate with a period at "p 16": Theory, “serves to summarize existing knowledge, to explain observed events and relationships, and to predict the occurrence of unobserved events and relationships” (Ary et al., 2010, p. 16), competence has describes an individual’s ability to complete a task (Cheetham & Chivers, 1996).

Also, insert a period after theory: "However, there has been a call for leadership research to continue to evolve beyond the dogma of existing theory,"

Include demographic information for the respondents in Results. Show and test for differences between Pilot 1 & 2 vs. Primary study participants. Also, how do the alumni in pilot 1 compare to current members in pilot 2?

Expand the footnotes for Figures 1 & 2 for readers not familiar with the model diagrams.

Comments on the Quality of English Language

Minor edits are needed.

Round 2

Reviewer 1 Report

Comments and Suggestions for Authors

The author had attended to most of the concerns raised in the original version of this manuscript. The editor can make final publication decision on this revised version. Thank you.

Comments on the Quality of English Language

Fine

Reviewer 3 Report

Comments and Suggestions for Authors

The author has revised the manuscript and it appears ready for publication.